# Effect of Gas Propellant Temperature on the Microstructure, Friction, and Wear Resistance of High-Pressure Cold Sprayed Zr702 Coatings on Al6061 Alloy

**Alessandro M. Ralls** [1] [ID], **Ashish K. Kasar** [1] [ID], **Mohammadreza Daroonparvar** [1,2,*], **Arpith Siddaiah** [1] [ID],
**Pankaj Kumar** [3,4] [ID], **Charles M. Kay** [2], **Manoranjan Misra** [3] **and Pradeep L. Menezes** [1,*] [ID]

[1] Department of Mechanical Engineering, University of Nevada Reno, Reno, NV 89557, USA;
alessandroralls@nevada.unr.edu (A.M.R.); akasar@nevada.unr.edu (A.K.K.); asiddaiah@nevada.unr.edu (A.S.)

[2] Research and Development Department, ABS Industries Inc., Barberton, OH 44203, USA;
charles.kay@hannecard.com

[3] Department of Materials and Chemical Engineering, University of Nevada Reno, Reno, NV 89557, USA;
pankaj@unm.edu (P.K.); misra@unr.edu (M.M.)

[4] Department of Mechanical Engineering, University of New Mexico, Albuquerque, NM 87131, USA

[*] Correspondence: mr.daroonparvar@yahoo.com (M.D.); pmenezes@unr.edu (P.L.M.)

**Abstract:** For the first time, Zr702 coatings were deposited onto an Al6061 alloy using a high-pressure cold spray (HPCS) system. In this work, five different $N_2$ process gas temperatures between 700 and 1100 °C were employed to understand the formation of cold sprayed (CS) Zr coatings and their feasibility for enhanced wear resistance. Results indicated that the $N_2$ processing gas temperature of about 1100 °C enabled a higher degree of particle thermal softening, which created a dense, robust, oxide- and defect-free Zr coating. Across all CS Zr coatings, there was a refinement of crystallinity, which was attributed to the severe localized plastic deformation of the powder particles. The enhanced thermal boost up zone at the inter-particle boundaries and decreased recoverable elastic strain were accountable for the inter-particle bonding of the coatings at higher process gas temperatures. The flattening ratio ($\varepsilon$) increased as a function of temperature, implying that there was a greater degree of plastic deformation at higher $N_2$ gas temperatures. The microhardness readings and wear volume of the coatings were also improved as a function of process gas temperature. In this work, the wear of the Al6061 alloy substrate was mainly plowing-based, whereas the Zr CS substrates demonstrated a gradual change of abrasive to adhesive wear. From our findings, the preparation of CS Zr coatings was a feasible method of enhancing the wear resistance of Al-based alloys.

**Keywords:** Zr702; cold spray; wear; surface modification; surface coatings

## 1. Introduction

Being one of the most widely used metals globally, aluminum (Al) has tremendously impacted an extensive array of industries spanning from aerospace to the automotive and medical sectors [1,2]. This is largely attributed to its impressive strength-to-weight ratio (with a density of ~2.7 g/cm³), high stiffness, and resistance to fatigue that many other metals do not offer [3]. Despite the widespread industrial use of Al, it often suffers significant drawbacks due to its relatively poor wear resistance and surface hardness [4]. As a consequence, many surface treatments have been proposed to control the inevitable wear of Al-based components. Such surface treatments that have been studied include but are not limited to laser shock peening, friction stir processing, and ultrasonic surface rolling process [5–8]. However, these techniques suffer drawbacks in the sense that they are only improving a pre-existing surface, which eventually will need to be replaced. Because of this, many have sought to use coating treatment methods as they are repeatable and can greatly extend the operational lifespans (especially in wear prone environments) of

Al-based components. Such methods include but are not limited to anodizing, physical vapor deposition, plasma electrolytic oxidation, powder sintering, additive laser surface treatments and chromium electroplating [9–17]. Through the utilization of these methods, the wear resistance of Al can be greatly improved, thus annually saving millions across a widespread of industries [18].

However, taking a microscopic view of these common surface coating treatments, they all in some form are either too expensive to use, emit too many harmful chemicals, or are simply not convenient [10,13]. From a metallurgical perspective, these techniques might also suffer from unwanted thermal diffusion, insufficient particle bonding and porous structures which can hinder the structural integrity of the coating consequentially worsening its wear resistance [17,19]. In place of this, many new technologies have been proposed in order to improve both the ease of deposition and performance of these coatings. One of these methods is known as the cold spray (CS) process [20]. In essence, this process relies on the extreme kinetic energies (typically with a particle velocity range of 300–1200 m/s) of powder particles (5–50 μm) launched at the desired surface. Upon impact, these powder particles then plastically deform due to the combination of adiabatic heating and high shear rates, the particles are then flattened in a jet-like formation and bonded to the substrate [20]. When compared to thermal-based coating techniques, the adhesion of CS coatings is superior in the sense that the oxide film on the surface of the substrate is eliminated from the high-speed impact of the particles [21–25]. In general, it is known that through higher particle velocities, the metallurgical bonding between the particle-to-substrate and particle-to-particle interfaces is enhanced, which can promote the formation of a dense coating (through layer-by-layer addition) with considerable adhesive and cohesive strengths [26–28].

Generally speaking, the bond strength of CS coatings has been reported to be exceptionally high across a variety of material substrates compared to other thermal-spray technologies. This is largely due to the compressive stresses that are generated from the mechanical interlocking and peening-like effects of the impacted particles, whereas tensile stresses occur from the particle heating of more thermal-based technologies [21,24,29]. Common material systems that have been reported to have sufficient bonding strength for CS coatings include but are not limited to metals (such as aluminum, titanium or nickel), their alloys (such as high-entropy alloys or Inconel), and various composites (such as metal matrix composites) [30–36]. For example, Chen et al. [37] found that by applying CS 316L and 316L-SiC coatings to AZ80 Mg-alloys (as a soft substrate) resulted with a very high bonding strengths being recorded at both $48 \pm 7$ and $53 \pm 9$ MPa. Similarly, Wei et al. [38] found that shot-peened assisted CS Ni coatings on AZ31B Mg-alloys yielded an impressive adhesion strength well above 65.4 MPa. In addition to this, Karthikeyan [39] has also reported that CS aluminum on Al6061 alloy can demonstrate a maximum bond strength of ~72 MPa, which demonstrates the bonding capabilities of softer metals, such as Al.

One industry that has been increasingly using CS coatings is nuclear power generation [40]. With Al being widely used in this industry, there is a growing need to preserve these components and extend their working lifespans [41]. Zirconium (Zr) 702 alloy is one metal that has earned a reputation of having a unique combination of exhibiting good wear and corrosion resistance at a large working range of temperatures [42]. One example of Zr coatings on Al in the nuclear industry can be found with high-performance reactor fuels. These coatings act as a diffusion barrier between the surrounding uranium, thus maintaining the tribological integrity of the coated fuel plates [43]. Similarly, other applications which Zr coatings can be greatly useful for can be found in aerospace and power generation due to the tribological robustness of Zr [44,45].

Considering the industrial usefulness of Zr coatings, there has been a minimal amount of research that has been studied on these coatings [46,47], let alone their tribological performances. In fact, from the available literature that has studied the effects of Zr coatings, there are no works to date that study the performance of Zr deposited by the CS process. Due to this, commercially pure (CP) Zr coatings on Al alloy was fabricated through a high-

pressure cold spray (HPCS) process. Through varying the $N_2$ gas processing temperature, the tribological behavior of these coatings was investigated. Understanding this, the novelty of this work lies from two different perspectives. First, it was discovered that Zr is indeed a suitable material that can be used for the CS process. This is largely due to the dense, robust, and oxide-free coatings achieved in this work. Second, the detailed microstructure and tribological mechanisms that are associated with Zr coatings was discovered with respect to varying CS processing parameters. Although the general mechanisms of CS have been increasingly known in recent years, these mechanisms tend to differ for varying material systems in conjunction with the standard processing parameters from CS (e.g., propellant gas type and processing gas temperature). For example, the work of Moridi et al. [48] determined that differences in materials crystal structures, physical properties, melting properties, and chemical reactions all drastically change the final product of CS coatings. This is quite impactful since the formation of the CS coating will determine its quality and performance in day-to-day applications. Because of this, each material system should be thoroughly studied in order to fully understand their behaviors from the deposition process to tribological testing. In this work, this was achieved with Zr where its structural quality and tribo-performance was studied as a function of propellant $N_2$ gas temperature. Through these findings, this work will contribute to the field of CS research and further the scientific understanding of Zr when subjected to the CS process.

## 2. Materials and Experimental Methods

In this work, CP Zr 702 powder with a particle size range of 20–45 µm was used for coating production, whereas the base substrate (abbreviated as S0) was a commercially available Al6061 alloy plate. Preparation of this plate was done through surface grinding with SiC abrasive paper (240 grit). The plate surface was then cleaned with alcohol and acetone before the CS process. Upon completion, an HPCS system (Impact Innovation 5/11 System, GmbH) supplied by ABS Industries (Barberton, OH, USA) was employed as the coatings to the Al-based alloy. Throughout the CS process, all CS parameters were held constant (Table 1) with the exception of the $N_2$ propellant gas temperature, which varied between 700 and 1100 °C. The abbreviations of the coatings in this work is set with respect to processing gas temperature. For specimens S1, S2, S3, S4, and S5, the operating gas temperatures were set to 700, 800, 900, 1000, and 1100 °C, respectively. Optical microscopy (IX70, Olympus, Tokyo, Japan) was used to characterize the polished-cross sectional microstructures of the as-sprayed coatings on the Al6061 alloy. In order to do this, the coated samples were cut, mounted, and polished to a mirror polish using a 0.05-µm low-viscosity alumina-slurry paste. After polishing, the thickness of each coating was measured to have a thickness of 600 ± 40 µm. Moreover, ImageJ software (version: 1.53.7) was employed to determine the porosity level of each coating (as per ASTM E2109–01) [49]. The structural phases of the coatings and powder were analyzed using a Bruker-D2 Phaser (Bruker, Madison, WI, USA) with Cu-K$\alpha$ radiation. The excitation voltage and current were set to 40 kV and 25 mA, respectively. The 2θ angles spanned from 20° to 100° with a scan speed of 1°/min. The step width was set to 0.02°. The scans of the CP Zr powder and Al6061 alloy substrate are shown in Figure 1a,b, respectively. For both specimens, only a single α phase was observed, indicating that there is no presence of other phases [50,51].

**Table 1.** Processing parameters of CS deposition.

| Propellant Gas | Sprayed Material | Gas Temperature (°C) | Gas Pressure (MPa) | Spray Angle (°) | Stand-off Distance (mm) | Step Size (mm) | Powder Reed Rate (RPM) | Powder Carrier Gas Flow Rate (m³/hr) | Type of Nozzle |
|---|---|---|---|---|---|---|---|---|---|
| $N_2$ | CP-Zr | 700–1100 | 3.0–5.0 | 90 | 25.4 | 0.5–0.1 | 1.0–2.5 | 2.5–3.5 | SiC Water Cooled Nozzle |

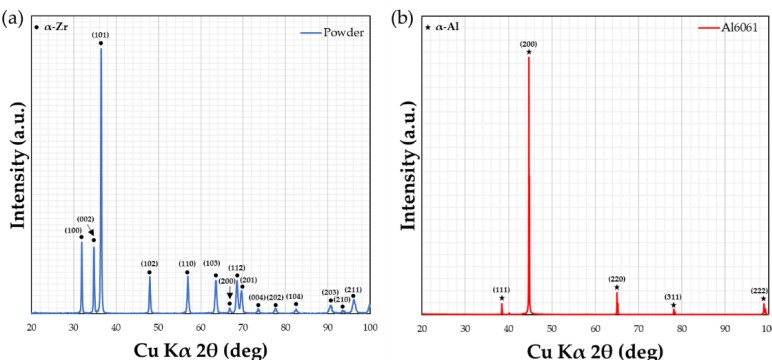

**Figure 1.** XRD patterns of the (**a**) Zr powder and (**b**) Al6061 used in this work as substrate.

Considering the variety of hardness testing methods, microhardness tests using a Vickers microhardness tester (Beuhler-Wilson, Tukon 1202, Binghamton, OH, USA) were administered to measure the microhardness of the base substrate and coating surfaces (after grinding and polishing). The primary reason for this was due to its non-destructive nature combined with its ability to produce repeatable and reliable readings, while preserving the coatings functionality after testing. For example, Rockwell hardness testing is one such method that can determine the hardness of coatings. However, due to its destructive nature, Rockwell hardness testing can destroy the usefulness of coatings after testing due to the debonding of various particles upon loading [52]. Similarly, nanoindentation is also one method that can be used for coatings, however due to the small indenting area, small defects, such as improper particle bonding or pores, can alter the hardness readings [53]. This can give a false representation of the coatings hardness, which would leave key information missing in the scientific analysis. Due to these facts, microhardness measurements were administered, similar to many other publications in CS and other thermal spray literatures [54–64]. In order to ensure repeatability, a total of 5 measurements under a 0.1 kgf were conducted and averaged.

For tribological evaluation, dry reciprocating tests (as per ASTM G133) [65] were performed using an Rtec multi-function Tribometer (Rtec-Instruments, San Jose, NY, USA) at room temperature (~25 °C). Before sliding tests, each sample was polished to an average surface roughness ($R_a$) of 0.1 ± 0.05 μm. An alumina ball with a 6.35-mm diameter was used as the counterpart during tribo-pair testing. For all reciprocating sliding tests, the applied loads varied from 5, 10, and 15 N, while the track length (10 mm) and velocity (2 mm/s) were held constant. A total distance of 1000 mm was chosen due to the stabilization of wear depth during testing. Afterward, the wear tracks of the samples were imaged and analyzed from the Rtec 3D optical profilometer (Rtec-Instruments, San Jose, CA, USA).

### 3. Results and Discussion

#### 3.1. Microstructure of the Coatings

Evaluating the microstructure of the coatings (Figures 2a–e and 3a–g), it can be seen that there is a clear local deformation of the impacted Zr powder particles. In a general sense, the impact of the Zr powder particles with high velocity can lead to severe plastic deformation, thus promoting a tight mechanical interlocking along the inter-particle boundaries [66]. This is quite evident from visually inspecting the bonding features between the coating to the Al6061 plate as well as the bonding between the deformed particles where the particles along all coatings exhibit a uniform flattened morphology. In fact, upon closer inspection, it can be seen that extruded lips from the impacted particles are present along the Zr-Al interface (as shown in Figure 4). In a sense, this phenomenon can be largely attributed to having an oxide-free interface, where the CS coating will have strong mechanical interlocking to the alloy substrate thus ensuring a high adhesion strength [33,38]. It is also worth mentioning that the subsequent particles from the CS process would also induce a peening-like effect along the interfacial layer, which would further the degree of deformation of the already deformed CS layer [67]. According to Xie et al. [68], the general

quality (i.e., the degree of metallurgical bonding) of CS coatings can be greatly improved from the strengthening effect of impacted/peened particles. If these subsequent particles are continuously impacted over time, a reduction in porosity and increase in adhesion strength can be observed. In the case of this work, this peening-like effect would be highly advantageous as it will allow for the already-sprayed Zr particles to further penetrate into the Al-based alloy substrate.

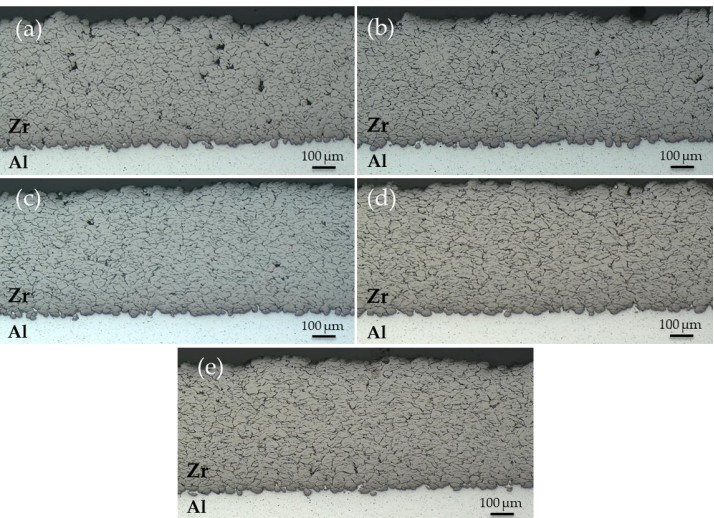

**Figure 2.** Optical micrographs of the polished and etched cross-section of samples (**a**) S1, (**b**) S2, (**c**) S3, (**d**) S4, and (**e**) S5.

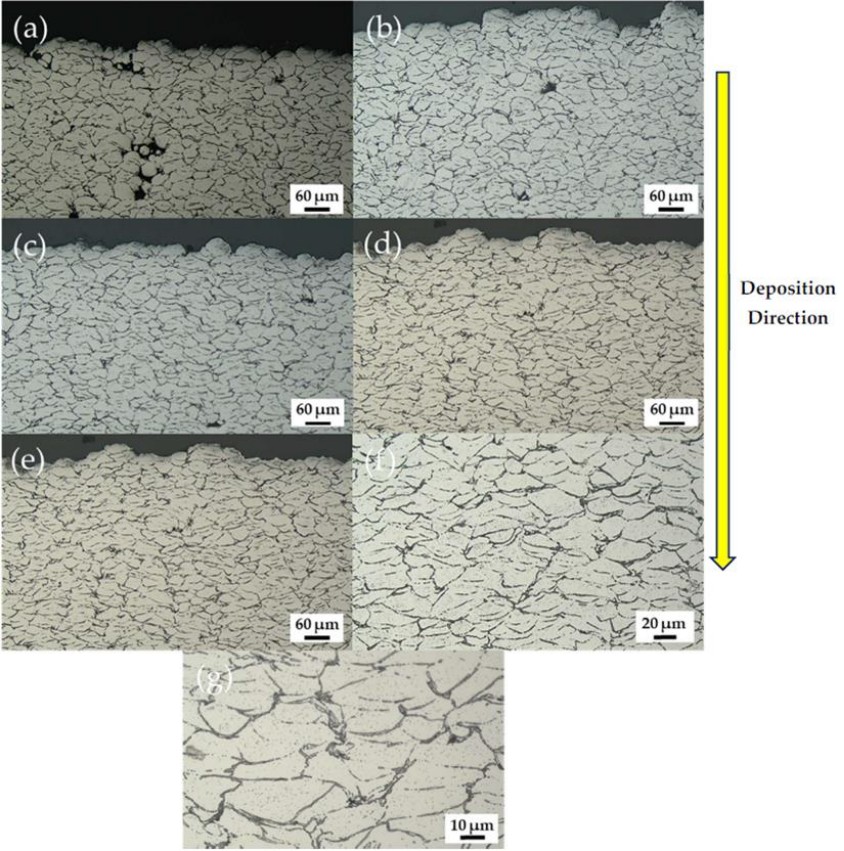

**Figure 3.** A higher magnification micrograph of the polished and etched cross-sections of (**a**) S1, (**b**) S2, (**c**) S3, (**d**) S4, and (**e**,**f**,**g**) S5.

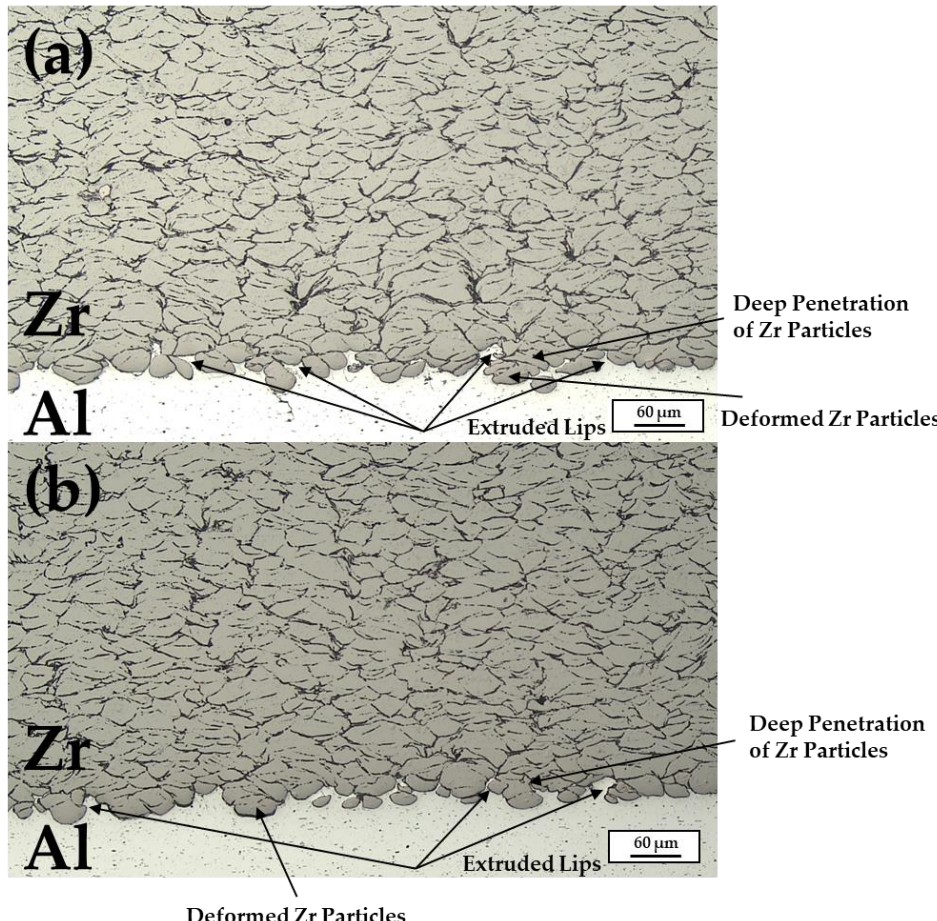

**Figure 4.** Two different views (**a,b**) of the formation of extruded lips from along the contacting interfaces of the CS Zr coating and the Al alloy substrate.

Upon closer inspection of Figure 4, it can also be seen that the degree of deformation from the Zr coating to the Al interface was measured to be 56 μm. When contrasted to the findings of other CS works, the observed degree of plastic deformation observed indicates that the CS coating is indeed of high quality [63,69,70]. For example, in the work of Sun et al. [63], similar findings of penetration depth were observed from CS Ti6Al4V and CoCrMo coatings on 6061-T651 Al alloy. In relation to their bonding strengths, it was found that Ti6Al4V coating demonstrated a bond strength of 50.38 MPa, whereas the CoCrMo coating measured at 66.17 MPa. With both bond strengths being quite impressive, this greater bond strength behavior of the CoCrMo coating was largely due to the decrease in porosity, higher hardness, and deeper penetration depth observed. Similar conclusions can be made from our findings.

With this being said, despite the coated Al-based alloys being sectioned, grounded, mounted, and polished for microstructural observations, both the coating as well as the Al alloy interface still demonstrated a high quality as there appeared to be no pores or microcracks along the layer-to-substrate and layer-to-layer interfaces (as previously shown in Figures 2–4). Having still been intact, it can be seen that the coating adherence (both between the contacting interfaces as well as the interlayers of the coating itself) are of exceptional quality. Considering that the particle velocities of the Zr particles are elevated from the general design of the HPCS process, it can be concluded that the combination of this phenomenon with the aforementioned reasons resulted in this finding.

To further expand on the coating quality and inter-particle boundaries, Yu et al. [71] has reported that the temperature generation along the interfacial regions of impacted particles (with gas pressures and temperatures of ~1 MPa and 800 °C) have reached values up to 576 °C above the initial gas temperature. Considering that the gas temperatures

and gas pressure in this work are far greater than what was observed by Yu et al. [71], it can be insinuated that the particles in this work were sufficiently deformed and bonded, as previously shown in Figures 2a–e and 3a–g [72]. To further support this, the melting temperature of Zr can be related to interfacial temperatures commonly known to promote high-quality particle bonding. For Zr, it is typically known that its standard melting temperature is ~1850 °C [73,74]. Understanding that peak interfacial temperatures can increase up to hundreds of degrees above the already heated particle temperature (influenced from the gas temperature), it can be assumed that the Zr powders would easily deform due to the enlarged strain fields of the particles [75]. According to Yin et al. [76], when interfacial temperatures exceed $0.4T_m$ (where $T_m$ is the materials melting temperature in K), enhanced dynamic recrystallization along the interparticle boundaries occurs. Given that these grains tend to be nano-sized, it has a positive influence to the coating integrity, thus acting as another mechanism for the coating bonding found in this work. Considering these earlier discussed points, the degree of plastic deformation of the coatings does seem to visually increase as a function of process gas temperature. In fact, the deformability of the particles can be assessed through a parameter commonly known as the flattening ratio ($\varepsilon$). The flattening ratio can be quantitatively expressed as [77]:

$$\varepsilon = \frac{w}{h}$$ (1)

where $w$ represents the width of the deformed particle and $h$ represents the height of the deformed particle. Through assessing this ratio, many important assessments can be made in regard to the structural integrity of the coating. As the flattening ratio increases, there is an implication of a greater value of cohesive strength along the bonded area [78]. In the case of CS Zr, the flattening ratio can be seen in Figure 5.

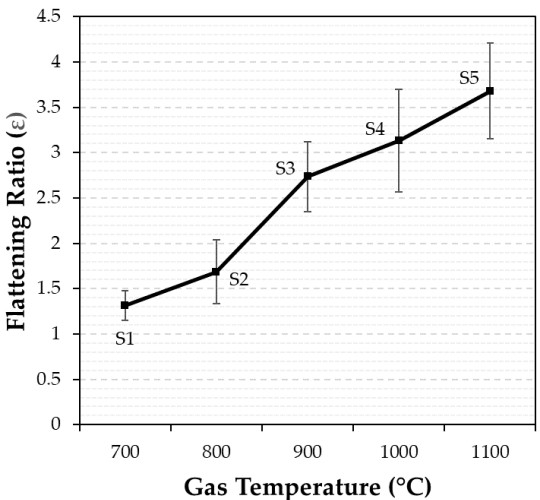

**Figure 5.** Evolutions of the flattening ratio, $\varepsilon$, of CS Zr with respect to processing temperature.

In Figure 4, the flattening ratio increases as a function of temperature, implying that there is a greater degree of plastic deformation at higher temperatures. It should be kept in mind that through gradually increasing the temperature, the velocity of the carrier gas is increased, thus increasing the local surface activation [79]. Typically, surface activation is dependent on the degree of localized plastic deformation, which creates a metallurgical bond of the impacted particles. Generally, sufficient bonding is achieved if the velocity of the particle is in between the window of its critical and erosion velocity. The process gas temperature produces a gas velocity, which can be expressed by the following equation [80].

$$v = \left( \frac{\gamma R T}{M_w} \right)^{\frac{1}{2}}$$ (2)



where $\gamma$ is the ratio of the constant pressure and the constant volume-specific heat that is normally 1.66 for monoatomic gases (e.g., helium) and 1.4 for diatomic gases (e.g., nitrogen and oxygen), respectively. *R* is the gas constant (8314 J/kmol·K), T is process gas temperature, and $M_w$ is the molecular weight of the gas. As mentioned earlier, when the powder particles with high velocity impact the substrate surface, the kinetic energy of particles turns into the mechanical deformation and thermal energy as well, which would explain the increase the degree of flattening from the Zr CS coatings. This would imply that the coatings are denser (which can also be visually seen) and would be able to resist early brittle fracturing, especially from tribological loadings [81]. Therefore, it can be insinuated from these findings that the main influence of temperature is to enhance the particle impact velocity which directly impacts the flattening ratio of CS Zr [80].

Taking a different perspective on the influence of the flattening ratio to the structural integrity of CS coatings, many have correlated these values to the coatings' bonding tensile strength. One recent work that has studied this is from Bagherifard et al. [82], in which the tensile strength of 316L SS coatings under various processing were correlated to the degree of particle flattening from the CS process. Under the processing conditions of a 5 MPa gas pressure and 1100 °C gas temperature, a flattening ratio of 0.36 ± 0.09 was observed. Although relatively lower than the findings of this work, the authors report that the CS deposit demonstrated nearly a 25% increase in ultimate tensile strength (UTS) compared to its bulk counterpart. Similarly, higher flattening ratios induced from altering processing gas temperatures (which similarly increases the particle impact temperature) for Ti have also showed noticeable improvement over its bulk counterpart, as reported by Binder et al. [83]. Therefore, it can be presumed that the incredibly high flattening ratio achieved would undoubtedly outperform bulk Zr, thus demonstrating the exceptional coating integrity found in this work.

Another important observation is the number of micro-defects present on the coating cross-sections. Reflecting back to Figures 2 and 3, specimen S5 led to the least number of microdefects among the deformed particles. When considering micro-defects, they are typically reported when there exists a number of micro-pores and micro-cracks, typically along the inter-particle boundaries [27,74] which can be seen with the lower temperature of $N_2$ process gas. Visually, S5 has the greatest resemblance of a pancake morphology with the greatest amount of equiaxed particles, which further supports the findings from Figure 4. Similar to the earlier findings, this can be attributed to the higher process gas temperature, which led to the production of a fully densified CP Zr coating on Al6061 alloy. It was noticed that the increasing impacting velocity increases the plastic flow of the impacting powder particles, leading to a considerable decrease in the defects/pores at the inter-particle boundary [84].

To better understand the relationship of porosity and gas temperature for Zr (having H.C.P crystal structure and lower slip systems compared to B.C.C and F.C.C crystal structures [85]), Figures 6a–e and 7 depict the visual and calculated porosities of the CS Zr coatings. Without etching, it is evident that the number of micro pores drastically decreases as a function of process gas temperature with the lowest recorded porosity at 0.16%. Given the explanation of the increased thermal softening of the powders particles during CS process, the gradual densification of the coating suggests that there will be an improvement in particle cohesion, which can imply a greater wear resistance as porosity defects can result in early crack propagation during sliding [86,87].

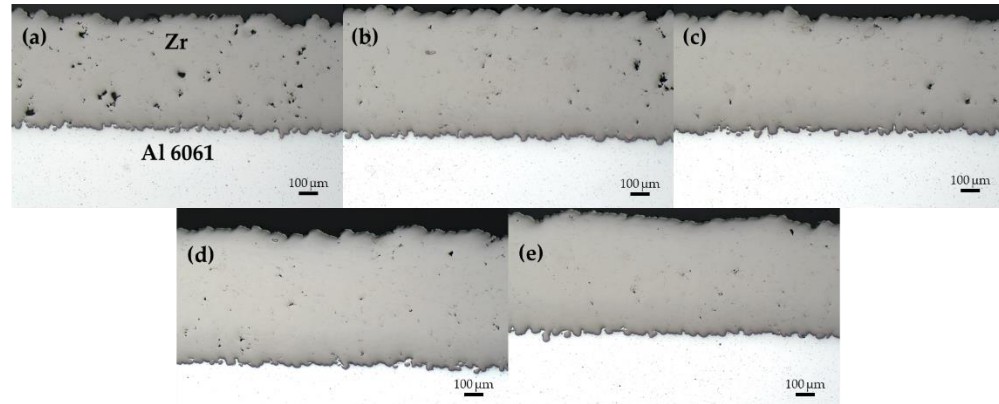

**Figure 6.** Micrographs of the polished cross-section of non-etched specimens (**a**) S1, (**b**) S2, (**c**) S3, (**d**) S4, and (**e**) S5.

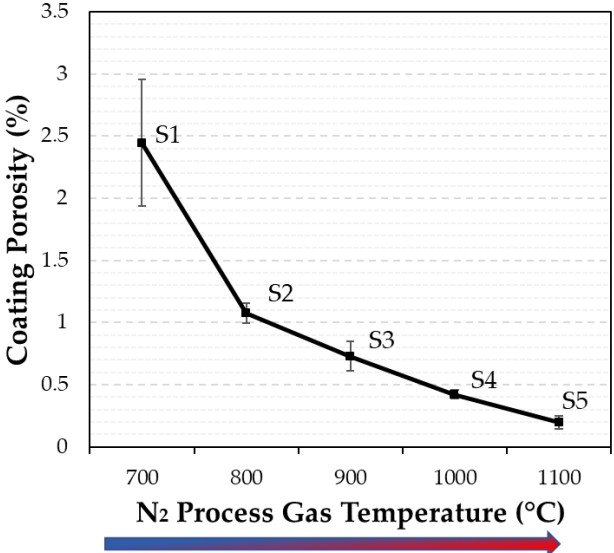

**Figure 7.** Porosities of the CS Zr coatings (S1–S5) with respect to $N_2$ process gas temperature (varied between 700−1100 °C).

Figure 8 depicts the XRD patterns of the as-sprayed coatings on Al6061 alloy substrate. Visually, there appears to be no evidence of either a phase transformation or oxidation with the coatings, which is typically seen from the CS process due to the hammering effects of the particles [88] and also low temperature deposition nature of the CS process [89]. For all specimens, the crystalline planes and phase structures also seem to be the same, primarily having a hexagonal close packed ($\alpha$) crystal system. Interestingly enough, there is a gradual decrease in peak intensity from S1 to S3, where afterward the peaks again begin to increase in intensity. This suggests that there is some initial peak broadening that occurs from the distorted atomic planes; however, from S4 to S5 the peaks begin to reduce in width suggesting that there is an increase in crystallite size [90]. To obtain a more visual understanding of the change with peak width, the full width at half maxima (FWHM) was calculated for each specimen based on the true peak broadening derived from the following equation [91]:

$$B = \sqrt{B_{obs}^2 - B_{inst}^2} \tag{3}$$

where $B_{obs}$ and $B_{inst}$ represent the observed peak broadening and the instrumental peak broadening. Based on this equation, the FWHM measurement for all specimens are represented in Figure 9. Based on these findings, it can be seen that the FWHM gradually increases for all samples after CS deposition, signifying that there is a refinement in crystal-

lite size and an increase in internal compressive stresses due to the hammering impact of the CS process [92].

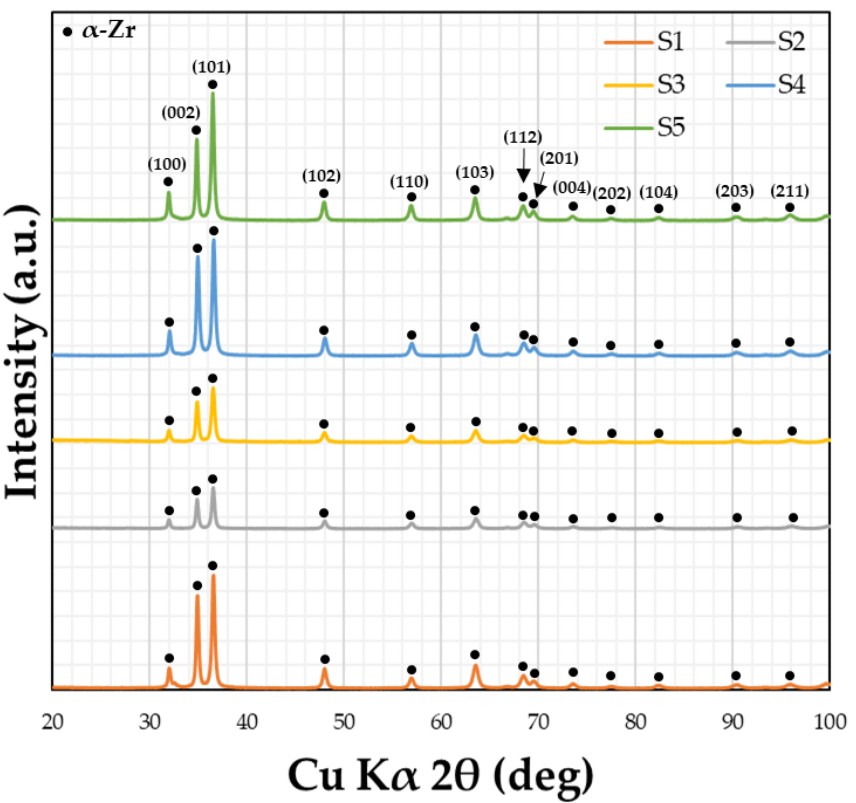

**Figure 8.** XRD patterns of the Zr CS coatings.

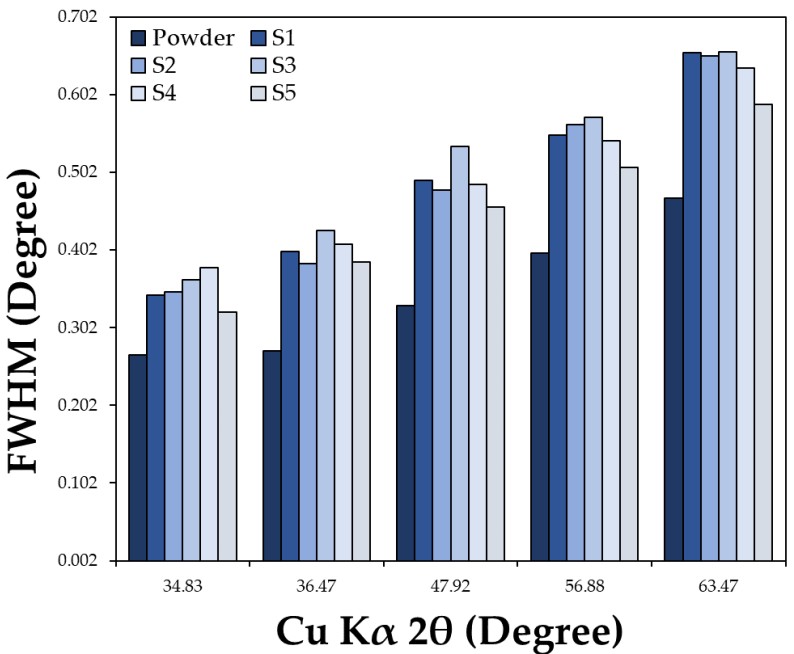

**Figure 9.** A comparative analysis of the calculated major FWHM peaks of CS Zr coatings.

For further understanding of this phenomenon, an analysis of the degree of plastic deformation to the coating must be measured. In order to measure this, the well-known Williamson-Hall (WH) equation was employed. In CS research, this equation has been

employed to measure the strain, crystallite size, and followed by dislocation density of fabricated cold sprayed coatings [93–96]. This equation is as followed:

$$\beta_\tau cos\theta = \varepsilon(4sin\theta) + \frac{K\lambda}{D} \tag{4}$$

where $D$ represents the crystallite size (nm), $\lambda$ is the radiation wavelength (A°), $\theta$ is the Bragg's angle (°), $K$ is the shape factor (defined at 0.9 for hexagonal close-packed (H.C.P.) crystal structures [97,98]), $\beta_\tau$ is the FWHM (radians), and $\varepsilon$ represents residual micro-strain. Based on each peak, the crystallite dimensions are calculated and averaged out which act as a representation for each sample. By comparing $\beta_\tau$ to the strain-induced broadening factor, $\beta_\varepsilon$, the slope can be computed, which allows for a quantifiable value of $\varepsilon$. Taking this value, the dislocation density, $\rho$, can be calculated by the following equation [91]:

$$\rho = \frac{2\sqrt{3}\varepsilon}{Db} \tag{5}$$

where $b$ is the Burgers vector magnitude for Zr. Based on these calculations, the WH plots for the CS powder and CS coatings are shown in Figure 10a–f, whereas the calculated crystallite size, dislocation density, and strain are shown in Figure 11a–c.

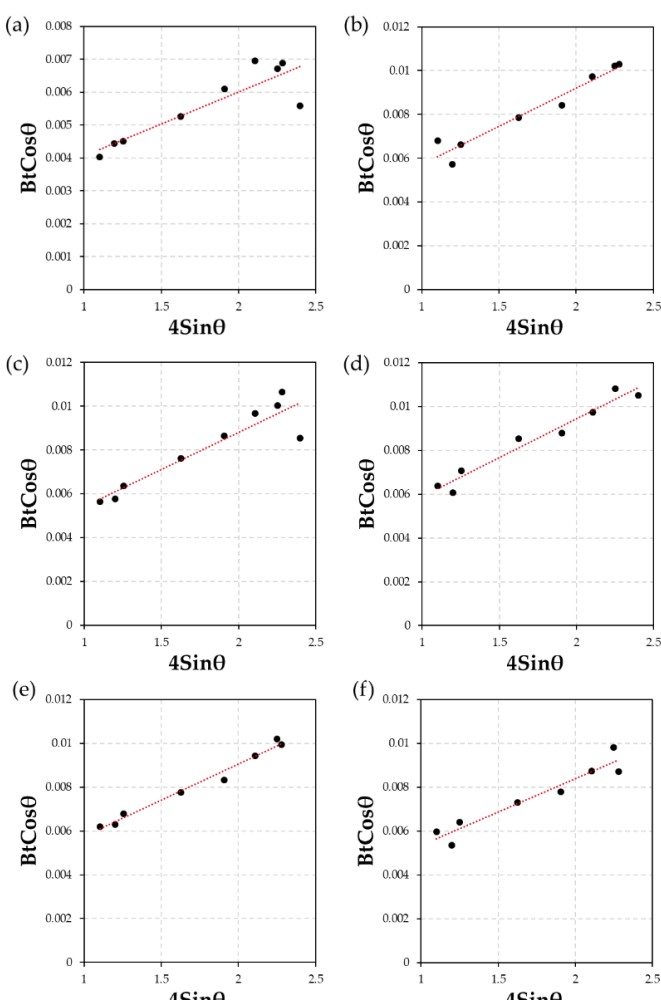

**Figure 10.** The Williamson-Hall plots for (**a**) the CS Zr powder and CS Zr coatings (**b**) S1, (**c**) S2, (**d**) S3, (**e**) S4, and (**f**) S5.

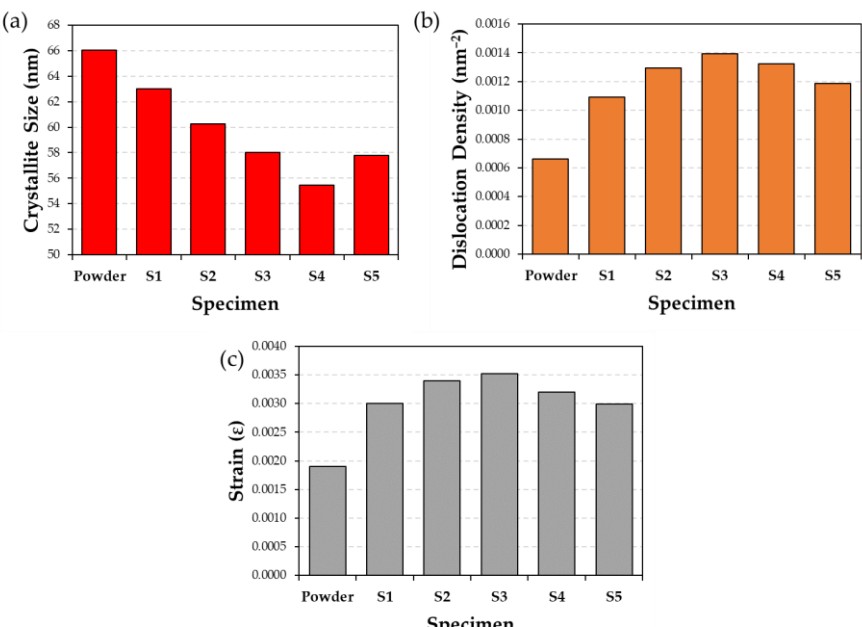

**Figure 11.** The average (**a**) crystallite size, (**b**) dislocation density, and (**c**) strain of CP Zr powder and as-sprayed Zr coatings (S1–S5).

According to these findings, there is a gradual decrease in crystallite size to 55.46 nm (S4) where it then increases to 57.77 nm for sample S5. In contrast to this finding, the maximum strain and dislocation density found for the tested samples were S3, where it then decreases again. To elucidate these findings, it is important to keep in mind the behavior of dislocations during the plastic deformation process. As reported by Liu et al. [99], often times both the generation and annihilation of dislocations throughout the CS process. Typically, this occurs when the dislocations generated will result with the previous dislocations traveling and decreasing the back stress on the activated source (deformed particle). A proposed model for this occurrence is described as:

$$\frac{d\rho}{d\gamma} = \frac{d\rho}{d\gamma}\Big|_{accumulation} - \frac{d\rho}{d\gamma}\Big|_{annihilation} = \frac{1}{\Lambda b} - \frac{y^*}{b}\rho \tag{6}$$

where $\lambda$ is the plastic strain from the CS process, $y^*$ represents the annihilation distance, $\Lambda$ is the mean free path, $\rho$ is the dislocation density, and b is the Burger vector.

As the process gas temperature increases, the kinetic energies of the particles are amplified, thus resulting in a higher degree of plastic deformation and increased dislocations. However, with the generation of new dislocations, there is also a more likely chance that dislocations will also be annihilated [100]. Considering that the temperature of the particles is directly increased, there is a greater chance of thermal diffusivity within the particle upon impact rather than heat transfer to the surrounding particles [101]. In the case that there is thermal diffusivity, defects in the form of stable dislocation loops reduce the propensity of localized recrystallization [102]. Thermal diffusivity in this case is very closely related to the small stresses released from the particle impact which can be close to the material melting point. From the rapid cooling of the impacted materials (Zr in this case) during this process, the excess in interstitial atoms and vacancies promote the growth of dislocation loops [99]. Above ~900 °C (process gas temperature used for S3), the strain and dislocation density decrease probably from the excess heat (thermal energy) thus increasing the crystallite size. However, it is important to note that the change in crystallite size is quite small and can be largely neglected as the densification of the Zr coating was shown to be greatly enhanced. Kumar et al. [84] showed multiple particle simulation images at 200 ns contact time for the different impact velocities. It was very conspicuous that the impact of Nb powder particles at higher velocities increases the jetted-out regions

in the CS Nb coating. This led to an increase in interface adiabatic temperature rise close to the melting point. Kumar et al. also noticed (in their simulation) the locations in which the interface temperature goes beyond the recrystallization temperature of Nb at around $0.4T_m$ ($T_m$: melting point of Nb). The enhanced thermal boost up zone at the inter-particle boundaries and also declined recoverable elastic strain energy were reported to be both accountable for the better inter-particle bonding in the coatings cold sprayed at higher process conditions (e.g., higher process gas temperature) [84,92]. Venkatesh et al. [103] depicted that the bonding process can be synergistically be influenced by the effect of velocity and temperature of the powder particles upon impact.

*3.2. Wear of the Bare and Coated Al6061 Alloys*

Traditionally, in order to assess the wear resistance of a material, the hardness must be measured as it is one of the major influencing factors of surface strength as per Archard's equation [104]:

$$Q = K\frac{PL}{H} \tag{7}$$

where $Q$ is the wear volume, $P$ is the normal applied load, $K$ is the sliding distance, and $H$ is the material hardness. It can be seen that the hardness is inversely proportional to the wear rate, implying that a higher hardness results with lessened wear.

The microhardness tests on the Zr coatings (Figure 12) show a gradual increase in hardness as a function of gas temperature, with the highest hardness being recorded at 482.6 HV. Compared to the base Al6061 plate, it is evident that all coated specimens exhibited a higher degree of mechanical hardness. Interestingly, specimen S5 showed the highest hardness despite the slight decrease in dislocation density. This can be largely explained due to the densification of the coating [105]. In the presence of voids, the structural integrity of the coating tends to suffer. Having a stronger inter-particle bonding at higher gas process temperatures, the amount of work hardening also increases, ensuring a sturdier surface [106]. Especially in the presence of pores and voids, there is a greater likelihood of unwanted crack propagations along the weaker regions of the pore due to increased stress concentration [107].

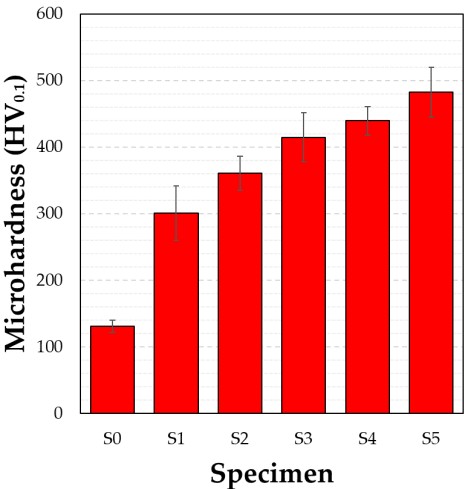

**Figure 12.** The average micro-hardness values of the bare and Zr-coated Al6061 alloys surface.

Applying the tribological loading to the Zr coatings, it can be seen that there is a decrease in friction with the Zr coatings in comparison to the bare Al6061 plate (Figure 13) which can be reflected from the decrease in crystallite size from the CS process [108]. However, these values are shown to vary, despite the change in load. These variations can be explained from the combination of the surface topography of the Zr coating and the counter ball material that was used. Given that there is some presence of porosity along the surface, there is a likelihood of brittle fracture, which likely generated some form of

third body wear that can cause instabilities with friction coefficient [81]. This in addition to the changing contact area of the ball can result in carrying concentrated pressures, which can fluctuate the frictional response from the system [109,110].

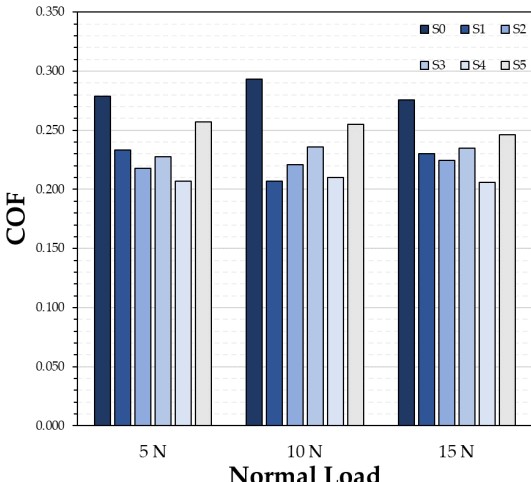

**Figure 13.** The change in frictional response for the bare and Zr-coated Al6061 plates as loads 5, 10, and 15 N.

A partial section of the wear track of specimens S0, S1, S3, and S5 are shown in Figures 14–17 alongside a two-dimensional profile of their respective wear tracks. It can be seen that the wear width and depth of the Zr-coated samples gradually decreases as the process gas temperature increases. This is to be expected due to the increase in hardness and as per Equation (7).

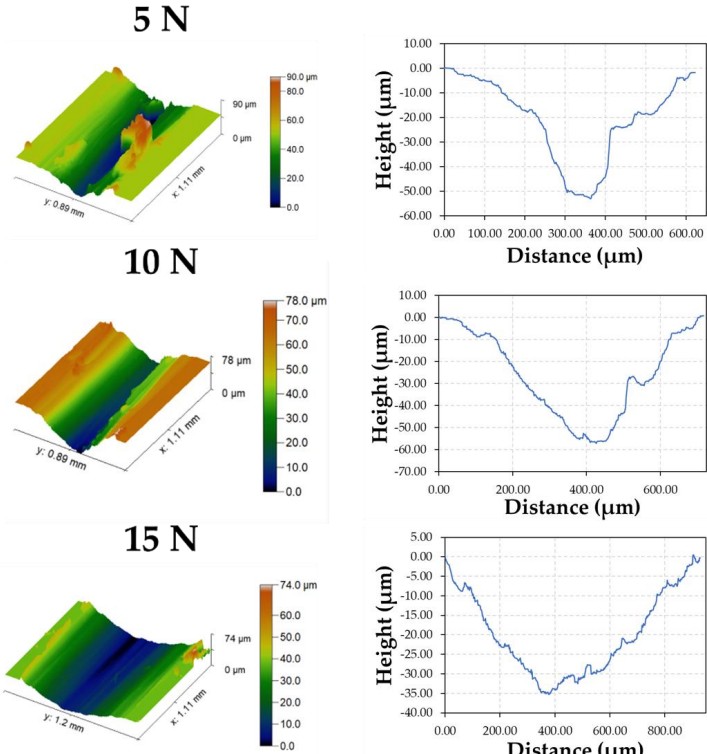

**Figure 14.** The wear tracks of the bare Al6061 plate under the normal loads of 5, 10, and 15 N.

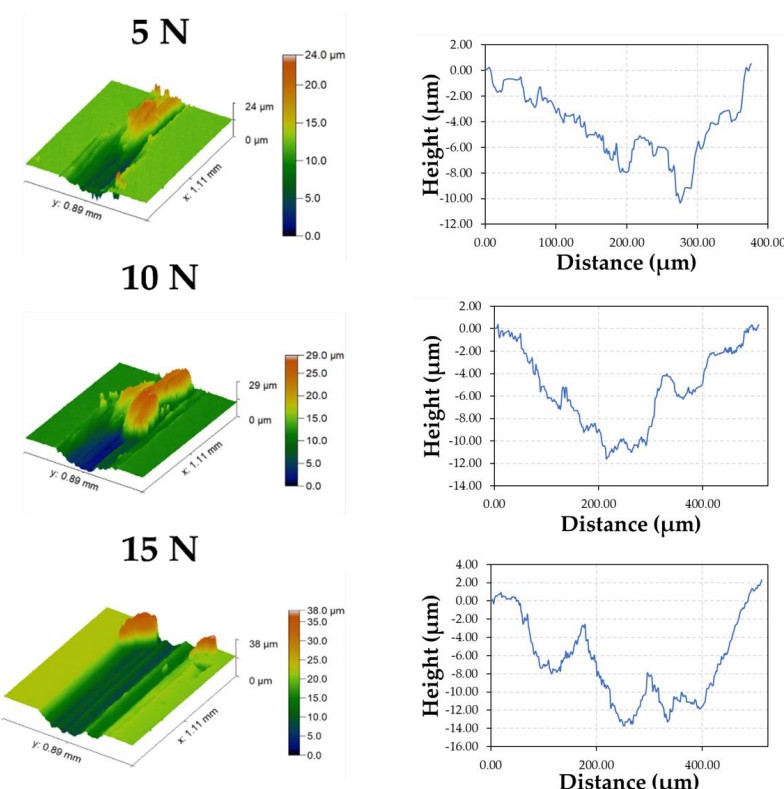

**Figure 15.** The wear tracks of the Zr-coated specimen S1 under the normal loads of 5, 10, and 15 N.

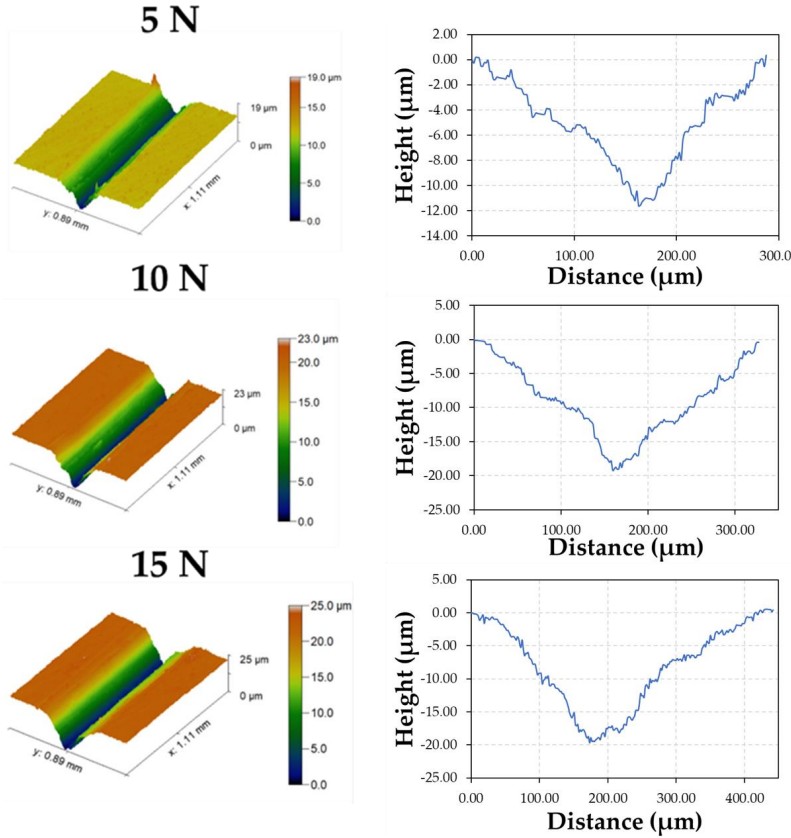

**Figure 16.** The wear tracks of the Zr coated specimen S3 under the normal loads of 5, 10, and 15 N.

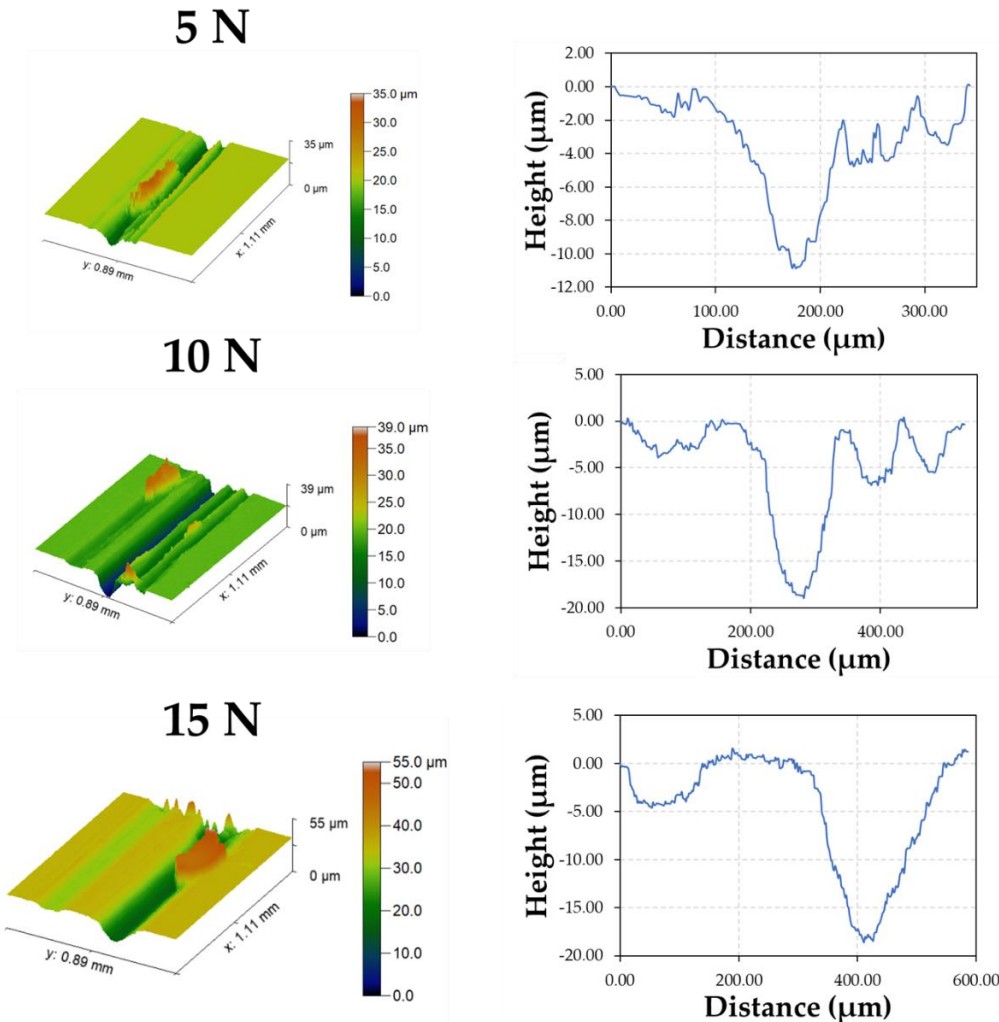

**Figure 17.** The wear tracks of the Zr-coated specimen S5 under the normal loads of 5, 10, and 15 N.

Further evaluating the wear track profiles, the wear mechanisms for all specimens largely changes. For the bare Al6061 alloy, plowing wear was found to be the dominant wear mechanism. This is to be expected as the material transfer from the Al6061 plate can influence work hardening of the transferred material promoting a greater degree of shearing from the asperity contacts [111]. Shifting the focus to specimen S1, there is a greater degree of sharper asperities that is present for all three tested loads. Their intensities gradually become sharper as the loads increase, insinuating a dominant abrasive wear mechanism. Considering that S1 had a greater amount of pores, the stress concentration at the point of tribological contact which can induce fatigue cracks from the influence of circular stress due to the brittle nature of CS coatings [31,86,112]. This increases the chance of enabling a fatigue wear mechanism of which brittle fractures occur from the buildup of microslips along the contacting asperities. This can increase the potential particle delamination, thus resulting with third-body wear to the system [104,113,114]. As the $N_2$ process gas temperature increases, the wear mechanism for resultant Zr coatings transitions to a dominant adhesive mechanism. This can be caused by the interactions of the asperities as the degree of surface energy can influence the localized adhesion strength. If the adhesions strength is greater than the breaking strength of the neighboring regions, adhesive wear occurs [104,113]. In the case of specimen S5, the wear track is non-symmetrical and, in some regions, does not have the appearance of material loss. To characterize the wear track, it appears to have parallel furrows, which have also been observed in other CS literature [115]. These results indicate that the localized stress of the coating (compared to S1) is decreased during tribological contact in addition to the increased hardness.

In order to have a better understanding of the influence of the $N_2$ process gas temperature to wear of resultant Zr coatings, the wear volume for the entire track was calculated and plotted, as shown in Figure 18. Similar to the wear tracks, the wear volume gradually decreases for each increasing sample (S1 to S5) with S5 having the smallest wear volume. These findings are consistent with the micro-hardness measurements as well as the dense and compact structure of S5. The higher process gas temperature enabled a full and efficient densification of the CS Zr coating which can promote a stronger bonding of Zr powder particles, thus increasing the wear resistance.

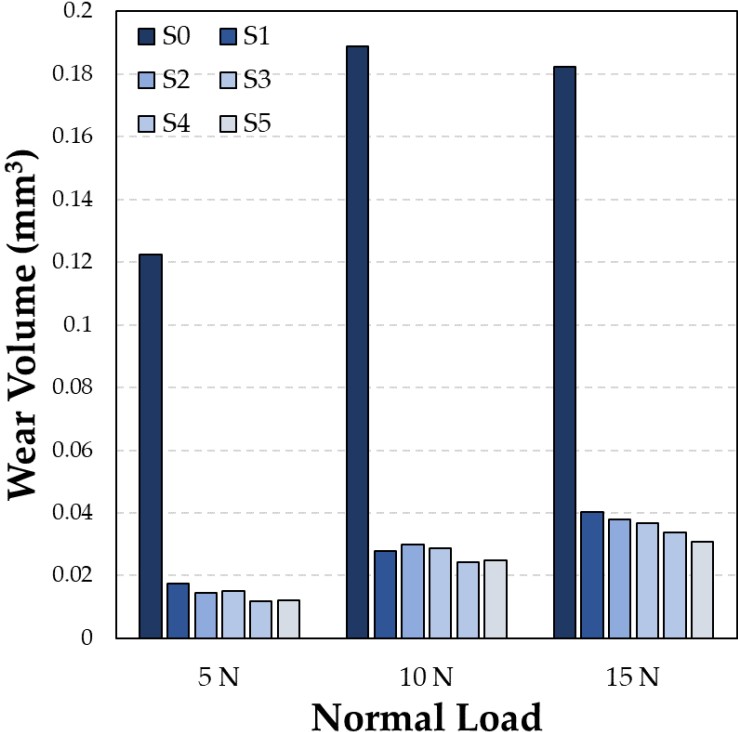

**Figure 18.** The wear volume of the bulk Al6061 substrate and CS Zr coatings.

## 4. Conclusions

In this work, the feasibility of novel CS Zr coatings on Al6061 substrates for enhanced tribological performance was studied. A total of 5 samples were fabricated with each sample having a gradual increase in $N_2$ process gas temperature from 700 to 1100 °C. The primary findings of this work are as follows:

1. The increase in process gas temperature allowed for a greater degree of thermal softening with the Zr particles, which helped create a dense and robust Zr coating.
2. Although there is no apparent phase transformation, the peak intensities from the XRD analysis drastically change as process gas temperature increased, indicating a refinement in crystallinity across all coatings.
3. The microhardness results indicate a gradual increase in hardness for each coating with respect to processing temperature.
4. The main wear mechanisms identified in this work are abrasive and adhesive. For all tested coatings, there was a gradual decrease in wear volume with respect to gas processing temperature.

Therefore, based on these findings we suggest that compact CS Zr coatings are an effective method of enhancing the wear resistance of commercially used Al6061 alloys.

**Author Contributions:** Conceptualization, A.M.R., M.D., P.K., C.M.K., M.M. and P.L.M.; Data curation, A.M.R., A.K.K., A.S. and P.K.; Formal analysis, A.M.R., A.K.K., A.S. and M.D.; Investigation, A.M.R., A.K.K. and A.S.; Methodology, A.M.R., A.K.K., A.S., M.D., P.K., C.M.K. and P.L.M.; Project administration, M.D., C.M.K., M.M. and P.L.M.; Resources, C.M.K.; Supervision, A.K.K., M.D., C.M.K. and P.L.M.; Validation, M.D., M.M. and P.L.M.; Visualization, P.K., C.M.K. and M.M.; Writing–original draft, A.M.R.; Writing–review & editing, A.M.R., A.K.K., M.D. and P.L.M. All authors have read and agreed to the published version of the manuscript.

**Funding:** This research received no external funding.

**Institutional Review Board Statement:** Not applicable.

**Informed Consent Statement:** Not applicable.

**Data Availability Statement:** Data sharing is not applicable to this article.

**Acknowledgments:** The authors would like to thank the National Science Foundation (CHE-1429768) for allowing the use of the powder x-ray diffractometer and ASB Industries, Inc., for providing the coated samples.

**Conflicts of Interest:** The authors declare no conflict of interest.

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
