# Peer review of "Effect of Gas Propellant Temperature on the Microstructure, Friction, and Wear Resistance of High-Pressure Cold Sprayed Zr702 Coatings on Al6061 Alloy"

_coatings, doi:10.3390/coatings12020263_

Round 1
Reviewer 1 Report
Additional information pieces are required for this article.
(1) The bonding types between the substrate and the coating and between deformed particles should be identified.
(2) Measurement of bonding tensile strength is necessary to quantify coating integrity.
(3) The term 'splat' may not be appropriate for CS coating unless the authors can prove that melting get involved with the process.
Reviewer 2 Report
Despite the efforts done by the authors, and the manuscript contains a good scientific story, there are some important points that should be considered by the authors.
- General Comments
- The authors used more than one form to express AA6061 Al alloy. One is Al 6061 alloy, and the other is Al6061 (without space) alloy. Please use one formula through the manuscript.
- The authors used separate parentheses to express references. A single parenthesis should be used with references, for example they used [Ref], [Ref] or [Ref] – [Ref]. Please change it to [Ref, Ref] or [Ref – Ref]. For example, page 1 - lines 31, 38, 43 etc.…
- There is repetition in some sentences or terms, check the full manuscript. For examples: Line 74 - the base substrate (abbreviated as S0) ….. Line 82 and 83 … on Al6061 substrates (abbreviated as S0), Line 80 … the exception of the N2 propellant gas temperature, which varied between 700 °C and 1100 °C …. With Line 81 …. the temperature of N2 process gas (chosen between 700°C - 1100°C)
- The authors used different forms to express 2θ angle (line 93 – o), (line 94 – degree) and (line 97 – Figure 1- deg). Please use one form through the manuscript.
- Introduction
- Lines 36 - 38: “Such methods include but are not limited to anodizing, physical vapor deposition (PVD), plasma electrolytic oxidation, additive laser surface treatments and chromium electroplating. Why you abbreviated only physical vapor deposition technique to PVD?
- Experimental
- Line 92, “The excitation voltage and current were set to 40 kV and 25 mA” please change it to “The excitation voltage and current were set to 40 kV and 25 mA, respectively)
- Clarify the coating conditions for the specimens from S1 to S5 related to the used N2 gas temperature.
- Please check Table 1 some words are divided in two lines such as (column 1 - raw 1) etc.….
- Results
- Lines 204 and 205: “signifying that there is a refinement in crystallite size and/or internal stresses due to the hammering impact of the CS process”. What it is the meaning of refinement in internal stresses?
- Conclusions
- The conclusions should be rewritten in specific points that contain the most important results without explanation

Reviewer 3 Report
In connection with a manuscript entitled; " Effect of Gas Propellant Temperature on the Microstructure, Friction and Wear Resistance of High Pressure Cold Sprayed Zr702 Coatings on Al6061 Alloy”, the authors did not make the necessary corrections as well as did not answer pressing questions regarding the preparation, and accordingly the major reviews are outlined which can be summarized as follows:
- The abstract should be written in the past tense with mentioning of the most prominent results obtained.
- It is from the authors to use these new references in the section on aluminum in the introduction:
- Rasha A. Youness 1, Mohammed A. Taha, Study of mechanical properties and wear behavior of nano-ZrO2-hardened Al2024 matrix composites prepared by stir cast method, J. Chem. Vol. 65, No. 2 pp. 307 - 313 (2022).
- Bezzina, E.B. Moustafa, M.A. Taha, Effects of metastable θ′ precipitates on the strengthening, wear and electrical behaviors of Al 2519-SiC/fly ash hybrid nanocomposites synthesized by powder metallurgy technique, Silicon (2022).
- Mahmoud F. Zawrah, Wafaa M. El-Meligy, Heba H. A. Saudi, Safae Ramadan, Mohammed A. Taha, Mechanical and Electrical Properties of Nano Al-Matrix Composites Reinforced with SiC and Prepared by Powder Metallurgy, Biointerface Research in Applied Chemistry 12(2) (2022) 2068-2083.
- S. AbuShanab, E.B. Moustafa, E. Ghandourah, M.A. Taha, The Effect of Different Fly Ash and Vanadium Carbide Contents on the Various Properties of Hypereutectic Al-Si Alloys-Based Hybrid Nanocomposites, Silicon (2021)
- The authors did not mention the novelty of their work. Thus, the novelty of this work should be clearly discussed.
- Was there a reaction between nitrogen gas and aluminum? The authors need to clarify this because I believe that there is an interaction.
- The hardness measurement is not completely accurate because it is known that the hardness measurement of a layer coating is done with rockwell hardness test, not vicker hardness test.
- The presence of errors in the references, such as the writing of the authors, etc., must be corrected

Round 2
Reviewer 3 Report
The authors carefully made all necessary adjustments. Therefore, I recommend that you accept the manuscript.
